# Supporting Parents Living in Disadvantaged Areas of Edinburgh to Create a Smoke-Free Home Using Nicotine Replacement Therapy (NRT): A Two-Phase Qualitative Study

**DOI:** 10.3390/ijerph17197305

**Published:** 2020-10-07

**Authors:** Rachel O’Donnell, Grace Lewis, Colin Lumsdaine, Giovanna Di Tano, Liz Swanston, Gillian Amos, Anne Finnie, Neneh Rowa-Dewar

**Affiliations:** 1Institute for Social Marketing and Health, University of Stirling, Scotland FK9 4LA, UK; 2School of Healthcare, University of Leeds, England LS2 9JT, UK; hcgml@leeds.ac.uk; 3NHS Lothian, Edinburgh, Scotland EH1 3EG, UK; Colin.Lumsdaine@nhslothian.scot.nhs.uk (C.L.); gditano@nhs.net (G.D.T.); Liz.Swanston@nhslothian.scot.nhs.uk (L.S.); gillian.amos@nhslothian.scot.nhs.uk (G.A.); anne.finnie@nhslothian.scot.nhs.uk (A.F.); 4Usher Institute, University of Edinburgh, Scotland EH8 9AG, UK; neneh.rowa-dewar@ed.ac.uk

**Keywords:** smoke-free home, nicotine-replacement therapy, tobacco smoke pollution, child, parents, qualitative research, vulnerable populations, pharmacies, public health

## Abstract

Exposure to second-hand smoke (SHS) in the home is largely associated with socio-economic disadvantage. Disadvantaged parents face specific challenges creating a smoke-free home, often caring for children in accommodation without access to outdoor garden space. Existing smoke-free home interventions largely fail to accommodate these constraints. Innovative approaches are required to address this inequality. In this two-phase study, we engaged with parents living in disadvantaged areas of Edinburgh, Scotland, to explore tailored approaches to creating a smoke-free home and develop and pilot-test an intervention based on their views and preferences. In Phase 1, qualitative interviews with 17 parents recruited from Early Years Centres explored alternative approaches to smoke-free home interventions. In Phase 2, an intervention based on parents’ views and preferences was pilot-tested with parents recruited through Early Years and Family Nurse Partnership centres. Seventeen parents took part in an interview to share their views/experiences of the intervention. Data from both study phases were thematically analysed. Phase 1 findings suggested that parents associated nicotine replacement therapy (NRT) with quit attempts but supported the idea of NRT use for temporary abstinence to create a smoke-free home, viewing this as a safer option than using e-cigarettes indoors. In Phase 2, 54 parents expressed an interest in accessing NRT to create a smoke-free home, 32 discussed NRT product choice during a home visit from a smoking adviser, and 20 collected their free NRT prescription from the pharmacy. NRT was used for up to 12 weeks in the home, with ongoing advice available from pharmacy staff. During qualitative interviews (*n* = 17), parents self-reported successfully creating a smoke-free home, quitting smoking, and reduced cigarette consumption, often exceeding their expectations regarding changes made. The intervention was acceptable to parents, but the multi-step process used to access NRT was cumbersome. Some participants were lost to this process. Parents living in disadvantaged circumstances may benefit from access to NRT for temporary abstinence in the home to assist them to protect their children from SHS exposure. Further research using a more streamlined approach to NRT access is required to determine the feasibility and cost-effectiveness of this approach.

## 1. Introduction

The harmful effects associated with children’s exposure to second-hand smoke (SHS) are well established [1,2]. Enabling parents to create a smoke-free home is key to the global reduction in children’s exposure to SHS. Recent work has shown that smokers who introduce smoke-free home rules are more likely to quit in the following six months [3,4]. Furthermore, children who live with smoking parents are more likely to become adult smokers themselves [5], but there is marginal evidence of an association of home smoking restrictions with reduced adolescent smoking behaviours [6].

Smoking in the home has increasingly become an inequalities issue, reflecting the significant challenges that disadvantaged parents face in creating a smoke-free home, particularly when sole caring for young children in accommodation with limited/no access to suitable and safe outdoor space, or ready access to outdoor space to smoke which enables parents to still monitor the safety of children who remain indoors. These challenges all constrain opportunities to smoke outside [7,8]. In Scotland, this is reflected in recent Scottish Health Survey (SHeS) data [9] published after the Scottish Government’s 2014 “Take it Right Outside” campaign [10], which aimed to raise awareness of the risks of smoking in the home. Data showed that the proportion of children exposed to second-hand smoke in their own home decreased from 12% to 6% in 2015, and it has remained in the range of 6–7% since [11]. However, analysis of the 2015 SHeS data highlights a clear social inequality in children’s exposure to SHS, with 12% of children exposed to SHS at home in the most deprived areas of Scotland, compared to less than 1% of children living in Scotland’s most affluent areas [12]. 

Despite reporting a high motivation to protect their children from SHS in the home, parents’ opportunities to act on health messages are often constrained by their domestic circumstances [13,14]. A range of interventions have been developed to promote smoke-free homes, for example using counselling approaches, feedback of a biological measure of children’s SHS exposure, school-based strategies, and educational materials. However, reviews of interventions to reduce SHS in households with children have concluded that there is insufficient evidence to recommend any specific approach [15,16].

Research and policy initiatives need to better acknowledge and accommodate the precarious and complex nature of vulnerable, disadvantaged parents’ lives and living conditions that make creating smoke-free homes difficult, to ensure that parents’ sense of hope, self-efficacy, and well-being are not compromised. On this basis, it has been suggested that temporary use of e-cigarettes or nicotine replacement therapy (NRT) could be offered to support parents who feel unable to quit and who struggle to create a smoke-free home, as refraining from smoking around children would reduce SHS exposure [14].

The National Institute for Health and Care Excellence (NICE) Harm Reduction Guidance [17] has identified the role of NRT for use in support of refraining from smoking temporarily (referred to from this point onwards as “temporary abstinence”) as a research gap. In Scotland, Health Scotland’s Harm Reduction Addendum [18] recommends that cessation services should advocate NRT for smokers for temporary abstinence to avoid exposing others to SHS, when smoking outside may be challenging. One qualitative study [19] asked disadvantaged caregivers their views on NRT to achieve temporary abstinence in the home. Most perceived NRT as being for personal reduction/cessation goals, and few linked the concept of NRT to temporary abstinence in the home, possibly reflecting a lack of knowledge about the health risks to children of SHS exposure. A more recent study supported caregivers in deprived communities in Nottingham to reduce their children’s SHS exposure through a 12-week intervention of behavioural support, NRT for temporary abstinence (provided by a specialist smoke-free home adviser through home visits) and feedback on SHS exposure levels using air quality monitoring [20]. Recruitment was labour intensive, but otherwise, this intervention was feasible and acceptable in supporting caregivers (*n* = 12) to reduce child SHS exposure levels. When intervention effectiveness was tested in a randomised controlled trial (RCT) (*n* = 205), significant decreases were found in SHS concentrations and cigarettes smoked in the home for the intervention group compared to a “usual care” group [21]. Cost-effectiveness was also demonstrated [22]. However, the specific effects of NRT for temporary abstinence remain unclear, based on the findings from this multi-component intervention.

Compared to the effects of SHS exposure, much less is known about the second-hand effects of e-cigarettes, and there is a lack of knowledge on the potential adverse health consequences to children and adults exposed to second-hand e-cigarette aerosols in the home [23]. Few studies have explored the use of e-cigarettes and their potential to reduce SHS related harms for children in the home. One recent study conducted in Scotland with 17 disadvantaged parents living in Edinburgh suggested that e-cigarettes were seen as potentially valuable in helping to protect children from SHS in the home when ability to smoke outside was constrained. However, parents raised concerns about e-cigarette safety, in particular related to child exposure to second-hand vapour, and concerns regarding their children playing with e-cigarettes in the home, which for some meant they would not consider using an e-cigarette in their home. Some parents raised concerns that using an e-cigarette in the home would expose children to the risk of a parental “smoking” role model, which participants recognised had been a contributory factor to them becoming smokers themselves [24].

Additional research is required on potential harm reduction options to enable health professionals to advise and support disadvantaged parents in making informed decisions on better protecting their children from SHS exposure in the home. In addition, approaches are required that not only acknowledge the limited opportunities that many disadvantaged parents have to take their smoking outside but provide a practical means to potentially overcome these circumstantial limitations.

On this basis, the aims of this two phase study were to:Explore disadvantaged parents’ views and preferences regarding harm reduction approaches to creating a smoke-free home (Phase 1).Pilot-test a smoke-free home intervention based on parents’ preferences to use NRT to create a smoke-free home (Phase 2).

## 2. Materials and Methods 

### 2.1. Phase 1 Methods

Parents were recruited through pre-arranged visits to four early years centres (EYCs) located in disadvantaged areas of Edinburgh, as categorised by the Scottish Index of Multiple Deprivation (SIMD) [25]. Parents are referred to EYCs by social services when considered vulnerable, often as a result of mental health challenges, drug or alcohol dependency issues, and/or family breakdown. In these circumstances, they receive free childcare and ongoing support from EYC staff and other agencies to improve their wellbeing and make positive health choices for their families.

Twenty parents, all of whom smoked, were provided with information sheets about the study, and eighteen expressed an interest in taking part in a face to face interview with one of the authors (RO), providing written informed consent to take part. Seventeen interviews were conducted within a designated private room in each EYC. One participant was uncontactable when the researcher attempted to schedule their interview.

The semi-structured interviews were audio-recorded with participant consent and lasted approximately 45 min. The topic guide covered: smoking history and previous quit attempts, current home smoking behaviours, previous experiences/use of NRT, and the appeal and applicability of using e-cigarettes and NRT for temporary abstinence to create a smoke-free home. Participants received a £15 supermarket voucher of their choice as a gesture of thanks for their time. Individual qualitative interviews and focus groups were also conducted with health and social care practitioners (e.g., GPs, pharmacists, health visitor, and EYC staff) (*n* = 15) and policy and practice leads with an SHS remit (*n* = 5) to ascertain their views on whether e-cigarettes and NRT are viable tools for reducing SHS in the home. These findings will be published separately.

Data collection and thematic analysis were conducted in parallel. The research team judged that they had reached data saturation, with no new themes generated, after 15 parent interviews, but the two additional interviews were conducted to affirm this. Interviews were transcribed verbatim, and transcripts were read and re-read to generate initial themes in line with Braun and Clarke’s [26] approach to thematic analysis, which is characterised by its flexible and iterative method for identifying, analysing, and reporting patterns as themes within the qualitative data. Two authors (RO, NRD) independently read each transcript, paying particular attention to the ways in which parents spoke about e-cigarettes and NRT, and how this related to smoking in the home. Discrepancies regarding theme or interpretation were discussed and resolved by these authors during a final comparative analysis, which focused specifically on parents’ views on e-cigarettes and NRT.

### 2.2. Phase 2 Methods

We aimed to recruit up to 20 disadvantaged parents who smoke and live in disadvantaged areas of Edinburgh and the Lothians, as categorised by the SIMD [24]. This number of participants was selected in order to include a diverse range of environments and personal contexts, whilst enabling data collection and analysis in the time available. Parents were recruited from EYCs and the Family Nurse Partnership (FNP). The FNP is a home visiting service for first time young mothers living in disadvantaged areas, who have a child under the age of 2 years. These mothers often live with other adult family members/partners, who the FNP also support. Recent data [27] suggest that on entry to FNP, 63% of mothers were perceived by their FNP nurse to have anxiety or other mental health issues, and 60% were living on low incomes. Edinburgh-based FNP data suggest that 62% of clients report smoking at some point during their pregnancy, and of these clients, 69% currently smoke [28].

Individuals were eligible to take part if they (1) smoked in the home, (2) cared for one or more children aged 5 years or under, at least once a week in the home. Exclusion criteria included individuals who were (1) currently pregnant or breastfeeding or (2) taking Warfarin, Clozapine, Theophylline, or Aminophylline, as NRT use in combination with any of these medications would require close monitoring from their GP. The study design is outlined in Figure 1:

EYC and FNP staff were contacted directly by the researchers in the first instance with information about the study. With agreement from EYC staff, two researchers (RO and NRD) visited to speak with parents about the study, providing them with information sheets and contact details for any follow-up questions or to register their interest in participating. The researchers also clarified that the focus of the study was on supporting parents to use NRT to create a smoke-free home, which was important as NRT is usually associated with quitting smoking. The study’s National Health Service (NHS) adviser (LS) then contacted each individual to arrange a suitable time to visit them either at home or at their EYC, to discuss study participation, obtain informed consent, and discuss NRT product choice. In the case of FNP clients, staff spoke with them about the study, at their discretion, during routine home visits, and a follow-up home visit with the NHS adviser was then scheduled with the parents who indicated an interest in taking part. With informed consent, parents were provided with an NRT recommendation letter to take to their local participating pharmacy, so that NRT for temporary abstinence in the home could be prescribed for up to 12 weeks.

Participants were asked to nominate the local pharmacy that they currently used, and any other nearby pharmacy that they were able to visit in case their first choice pharmacy was unable to participate in the study. The researchers then made contact with pharmacy managers to discuss their participation and the protocol for prescribing and dispensing NRT products for temporary abstinence in the home. Study information sheets were shared with pharmacy staff, who completed NRT product tracker sheets when individuals were prescribed NRT so that product type and quantity prescribed were logged.

Qualitative interviews were conducted with parents to capture their perspectives and experiences of taking part in the pilot intervention. Interviews with parents took place once they had stopped using NRT for temporary abstinence, up to 12 weeks after their initial pharmacy prescription. Parents were also invited to take part in an interview if they had (a) decided against trying NRT after it had been prescribed; (b) stopped using the NRT product before the prescription period ended; or (c) quit smoking before the end of the prescription period. The topic guide covered: participant background, smoking history, household structure, home smoking patterns pre and post NRT use, experiences of engaging with the NHS adviser and pharmacy staff, and the acceptability of the intervention as a whole. At the end of each interview, participants were offered a £15 supermarket voucher of their choice as a gesture of thanks for taking part. Individual qualitative interviews (*n* = 7) were also conducted with FNP and pharmacy staff, and with the NHS adviser to ascertain their views on the feasibility and acceptability of this approach. These findings will be published separately. All interviews were digitally recorded with participant consent and transcribed verbatim for in-depth analysis. Data collection and analysis took place concurrently, and data were analysed thematically using the approach already outlined within the Phase 1 Methods Section.

## 3. Results

In the Phase 1 and Phase 2 results that follow, parents’ names are presented as pseudonyms throughout to protect their anonymity.

### 3.1. Phase 1 Results

Our sample consisted of 16 mothers and one father, whose ages ranged between 20 and 45 years of age. All of the mothers interviewed were unemployed, and the father worked part-time. Thirteen participants were separated, and four lived with their partner. All participants had at least one child aged five years or under. Five participants reported that they lived in a smoke-free home, and 12 restricted home smoking to specific rooms (for example, the living room and/or the kitchen) to try to reduce child exposure to SHS in the home. In the illustrative quotes that follow, parents’ pseudonyms are presented alongside their home smoking status.

#### 3.1.1. Parents’ Views on Using e-Cigarettes to Create a Smoke-free Home

Although a few parents expressed initial enthusiasm for using e-cigarettes to create a smoke-free home, most parents raised specific concerns regarding the perceived safety of e-cigarette use around children, in particular related to children’s exposure to second-hand aerosols from e-cigarette use indoors (Figure 2, quote 1).

Some parents had already established e-cigarette home-smoking rules for this reason and sent visitors to the home outside to “smoke” them on this basis (Figure 2, quote 2). Parents also raised concerns about the potential appeal of e-cigarettes to children, based largely on product design, which could encourage children to pick them up and explore them if they were left lying around (Figure 2, quote 3). In addition, some parents spoke of the challenges they might face in affording what they perceived to be a good quality, reliable e-cigarette device and accessories in the first instance, which was seen as another barrier to using them for temporary abstinence to create a smoke-free home (Figure 2, quote 4).

#### 3.1.2. Parents’ Views on Using NRT to Create a Smoke-Free Home

Parents associated NRT specifically with quit attempts, and several had used patches, gum, and/or lozenges previously to assist with stopping smoking. However, parents were generally positive about the potential use of NRT products for temporary abstinence in the home (Figure 2, quote 5). Parents who had unsuccessfully used NRT products to support a previous quit attempt were also open to using NRT for temporary abstinence and were not put off by their previous experiences, given the range of NRT products available for use (Figure 2, quote 6). In particular, the NRT inhalator was viewed positively by some parents as it enabled the hand to mouth actions associated with smoking, but without any SHS exposure in the home, or the second-hand aerosols associated with e-cigarette use (Figure 2, quote 7). Only one participant raised concerns regarding the financial cost associated with NRT use, specifically related to quitting in the first instance. However, in relation to the use of NRT for temporary abstinence to create a smoke-free home, they suggested that they might be willing to pay for NRT depending on the cost (Figure 2, quote 8). 

Several issues became clear as a result of these interviews. Firstly, our findings reinforced the need to find workable solutions for parents living in disadvantaged circumstances, who were keen to explore new options and wanted to adequately protect their children from SHS exposure in the home. Secondly, whilst parents were initially receptive to the use of e-cigarettes for temporary abstinence, they identified several potential barriers associated with using them to create a smoke-free home, most notably related to the perceived safety of use around children in the home. These findings support previous research already outlined exploring the notional use of e-cigarettes to create a smoke-free home [24]. Concerns based on emerging evidence have recently been published, suggesting ways in which e-cigarettes could introduce health-related risks in relation to early-life exposures, including in instances where parents wrongly perceive that e-cigarettes do not produce second-hand aerosols, and given that nicotine from e-cigarettes can deposit in low amounts on surfaces, suggesting risks to children associated with third-hand smoke exposure [29]. On this basis, we recognised that there was limited value in developing an intervention that introduced e-cigarettes as a potential solution to creating a smoke-free home. The use of NRT products for temporary abstinence in the home, on the other hand, was viewed as having the potential to provide a tailored solution for parents/carers living in disadvantaged areas given their specific challenges associated with creating a smoke-free home. On this basis, in Phase 2, our intervention explored whether NRT provision for use in the home supported parents to better protect children from SHS in socioeconomically disadvantaged homes.

### 3.2. Phase 2 Results

Fifty-four parents expressed an interest in taking part in the study; 32 met with the NHS adviser; and 20 were prescribed NRT. Five participants who showed an initial interest in the study did not meet the study inclusion criteria. Additional barriers to participation included moving out of the area and the onset of stressful life events.

Seventeen parents (15 who were prescribed NRT, and two who chose not to visit their local pharmacy to access their free NRT supply) took part in a face to face qualitative interview to explore their experiences of taking part in the study. This included two sets of paired interviews, conducted at the participants’ request. Three participants who had collected their NRT prescription were lost to follow-up. Changes in numbers of cigarettes smoked and/or smoking location were evident in most participants’ accounts, as outlined in Table 1.

Two participants did not collect their NRT prescription, citing previous negative interactions with pharmacy staff as a barrier to participation. One participant reported no change to their smoking practices. Two reported quitting smoking after NRT use, six reported creating a smoke-free home, and 12 reported reduced cigarette consumption, in some cases by 50% or more. In the illustrative quotes that follow, parents’ pseudonyms are presented alongside any reported changes made to their smoking behaviour after NRT use for temporary abstinence in the home.

#### 3.2.1. Acceptability and Feasibility of NRT Use for Temporary Abstinence to Create a Smoke-Free Home 

Some participants spoke of ways in which they had successfully broken existing smoking habits to accommodate NRT use in the home for temporary abstinence, in particular related to their first cigarette of the day (Figure 3, quote 1).

Participants also spoke of needing time and willpower to use and maintain use of NRT products, which helped them to go on to establish new home smoking behaviours (Figure 3, quote 2). During this initial transition period, some participants switched NRT product in discussion with pharmacy staff, recognising that their first choice product did not suit them (Figure 3, quote 3). However, one participant did not recall knowing that they could switch NRT product if they found their initial choice unsuitable. Unable to maintain NRT use with a product they considered to be suboptimal, they made no subsequent changes to their home smoking behaviour.

There were some challenges associated with collecting NRT prescriptions from pharmacies, which sometimes hindered successful access to NRT. These included the impracticalities and inconvenience associated with what was perceived to be long waiting times, and the lack of privacy that open pharmacy consultations entailed. Poor existing relationships between participants and pharmacy staff also served as a barrier, as was the case for two participants, who were distant relatives, who reported having negative interactions with pharmacy staff in the past (Figure 3, quote 4). These findings are presented and discussed elsewhere [30]. However, in most cases, participants reported more positive, supportive interactions with pharmacy staff.

#### 3.2.2. Unanticipated Benefits Associated with NRT Use

Participants reported a number of unintended positive consequences associated with NRT use for temporary abstinence in the home, including financial savings, and the ability to spend more time with their children (Figure 3, quotes 5 and 6). Several participants suggested they were put off from trying to make smoking behaviour changes when the stated end goal for health professionals was complete cessation (Figure 3, quote 7). However, participants often exceeded their own expectations of their ability to make smoking behaviour changes using NRT for temporary abstinence indoors, needing minimal or no support from EYC and FNP staff, and sometimes this gave them new-found confidence to believe that they might quit smoking altogether. In part, their exceeding of expectations may have been because of the initial focus on creating a smoke-free home, which might at this stage feel like a more attainable goal for some parents than quitting smoking.

## 4. Discussion

The findings of this two-phase study suggested that providing disadvantaged parents with free NRT for temporary abstinence in the home was both feasible and acceptable, supporting them to reduce children’s exposure to SHS in the home through the creation of a smoke-free home, via reduced smoking consumption, and in a minority of cases, as a result of quitting smoking. Parents often exceeded their own expectations of their ability to change their smoking behaviours, with minimal or no support from EYC/FNP staff. They also reported unanticipated benefits associated with home smoking behaviour changes, including more time spent with children, financial savings related to reduced smoking, and a newfound belief that they may be able to quit altogether.

However, the multi-step process used to access NRT was cumbersome, and some participants were lost during this process; only 20 of the 54 participants who expressed an interest in taking part went on to obtain their NRT prescription. Retention rates could be improved by streamlining this process so that parents discuss NRT product choice with their local community pharmacy from the outset, eliminating the need for a specific home visit by a trained adviser. This approach could be more appealing and less burdensome to participants, and it aligns well with recent calls to improve the use of community pharmacists as the first port of call for health-related advice [31]. Pharmacists have existing expertise and experience of providing NRT advice (in relation to smoking cessation), and adapting service provision to use of telephone consultations, for example, could overcome the challenges observed in this study associated with open pharmacy discussions. In Scotland, telephone and video consultations are already being used as alternative approaches to face-to-face consultations in some instances, including most recently to reduce exposure risks associated with the Covid-19 pandemic. As community pharmacies are freely accessible to communities, involving local pharmacy staff directly in discussion with parents about NRT product choice could be more practical and sustainable in the longer-term. In Scotland, there are over twice as many pharmacies in the most deprived areas as in the least deprived areas [32], and they also deliver smoking cessation services so would be well placed to support parents. Supportive encouragement in non-judgemental environments is key to support parents to create a smoke-free home without perpetuating stigma [30]. To our knowledge, engaging community pharmacies more directly in providing NRT for temporary abstinence to create a smoke-free home has yet to be explored, and this approach warrants further research. 

Previous research [20] has suggested that smoke-free homes research is labour intensive, and that only a small proportion of potential participants report smoking in the home to begin with. Initially, our recruitment experience supported this suggestion, with parents often stating that they did not smoke in the home, which was sometimes contrary to EYC and FNP staff perceptions of their home smoking behaviour. In an attempt to overcome this, and recognising the potential guilt and stigma often associated with smoking in the home as a parent [14], recruitment questions were adapted from “Do you ever smoke in the home?” to “Are there ever times you find it difficult to keep your smoking outside of the home, such as at night time, during bad weather or when your child is unwell?” This strategy led to more open initial interactions with parents, who may have perceived this to be a less judgemental approach. Future research could adopt a similar strategy during recruitment to ensure that potential participants are not further stigmatised for their home smoking behaviour or alienated from potential sources of support, especially given that parents generally strive to do the best they can, in often challenging circumstances [7]. 

A few participants informally shared their NRT with their partner at home, during the course of the study. Whilst this was not formally encouraged, this approach fits with recent calls for future smoke-free home interventions to be delivered at household level, rather than being largely mother-led [33,34]. Developing smoke-free home interventions that work with men, women, and other family members on an equal basis could better frame household smoking as a collective responsibility, and so take the pressure off women as mothers and wives to persuade others to take their smoking outside [13]. The use of NRT for temporary abstinence in the home could represent one means of developing a household level smoke-free home intervention in the future.

There were a number of limitations to our study. Interviews were only conducted with two participants who did not visit their local pharmacy to obtain their NRT prescription in Phase 2, so little is known about the barriers to accessing NRT in this context, although findings suggest that positive relationships with pharmacy staff are key. The positive accounts of parents who were interviewed may have been influenced by social desirability, although their outcomes were often informally validated by EYC/FNP workers. However, measuring home SHS levels (e.g., using devices to measure fine particulate matter (PM_2.5_)) as a well-established proxy measurement for SHS in indoor air [35] at baseline and follow-up would provide an objective measure of smoke-free home success in future. Introducing a follow-up component would also enable verification of longer-term behavioural outcomes, which would be valuable given that recent work has shown that smokers who introduce smoke-free home rules are more likely to quit in the following six months [3,4]. In addition, whilst parents were given access to a free 12-week supply of NRT, they generally only visited the pharmacy 2–3 times, suggesting that this could be a lower cost approach than anticipated. An assessment of the cost effectiveness of this approach is therefore required. 

A particular strength of this study over previous work was that it not only acknowledged the limited opportunities that disadvantaged parents have to take their smoking outside, but the study also provided a practical means to potentially overcome these limitations, through access to free NRT for temporary abstinence in the home. This approach, which was developed as a direct response to the views and opinions of parents who were interviewed in our Phase 1 study, also built on the notion that parents are motivated to protect their children from SHS in the home and doing their best as parents in difficult circumstances. It has been suggested that there is currently an overreliance on strategies which focus on negative reinforcement, specifically in relation to attempts to change smoking norms and increase smoke-free public spaces [36]. We argue that the same applies to smoke-free home interventions, which are usually targeted at mothers living in disadvantage [7,12], and often focus on highlighting the harms associated with childhood exposure to SHS in the home, and the need to take smoking outside, without providing a viable solution to enable this change.

## 5. Conclusions

The findings of this two-phase study emphasise the need for tailored approaches to smoke-free home interventions that acknowledge the limited opportunities that parents living in disadvantaged communities have to take their smoking outside. The use of free NRT for temporary abstinence in the home could provide a potential solution to the complex challenges that disadvantaged parents face in trying to create a smoke-free home. Parents considered this approach both feasible and acceptable, and it supported them to reduce their children’s exposure to SHS in the home, through creation of a smoke-free home, via reduced cigarette smoking, and in a minority of cases, as a result of quitting smoking. As community pharmacies are more often located in disadvantaged communities and are freely accessible, involving them in the delivery of this intervention could be practical and sustainable in the longer term. Further research is required, incorporating a more streamlined approach to NRT access, to test the feasibility and cost-effectiveness of this approach.

## Figures and Tables

**Figure 1 ijerph-17-07305-f001:**
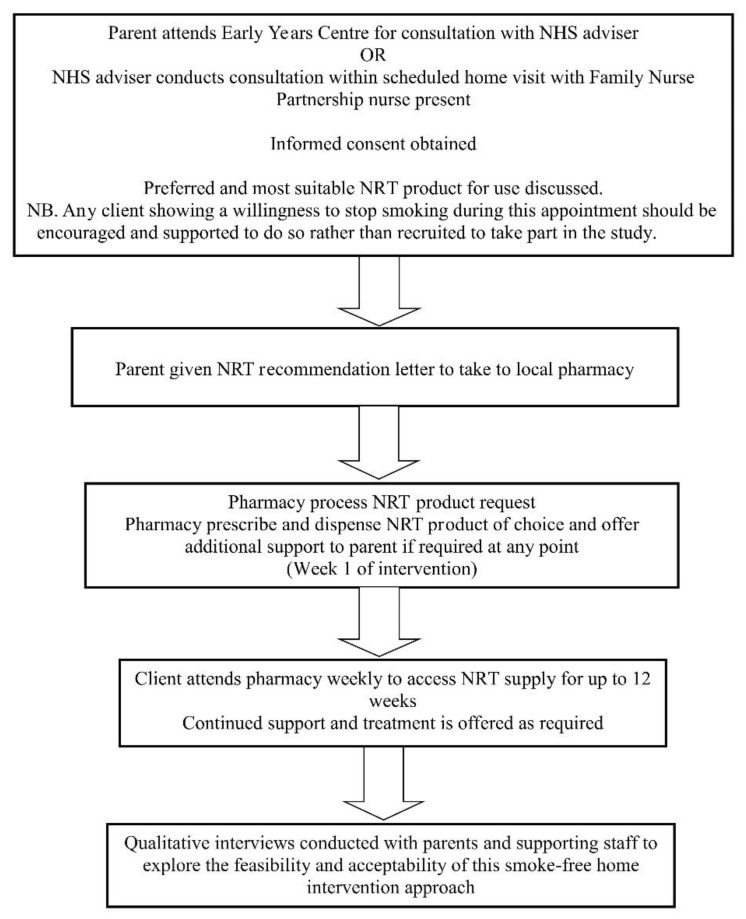
Phase 2 study design. Abbreviations: NHS—National Health Service; NRT—Nicotine replacement therapy.

**Figure 2 ijerph-17-07305-f002:**
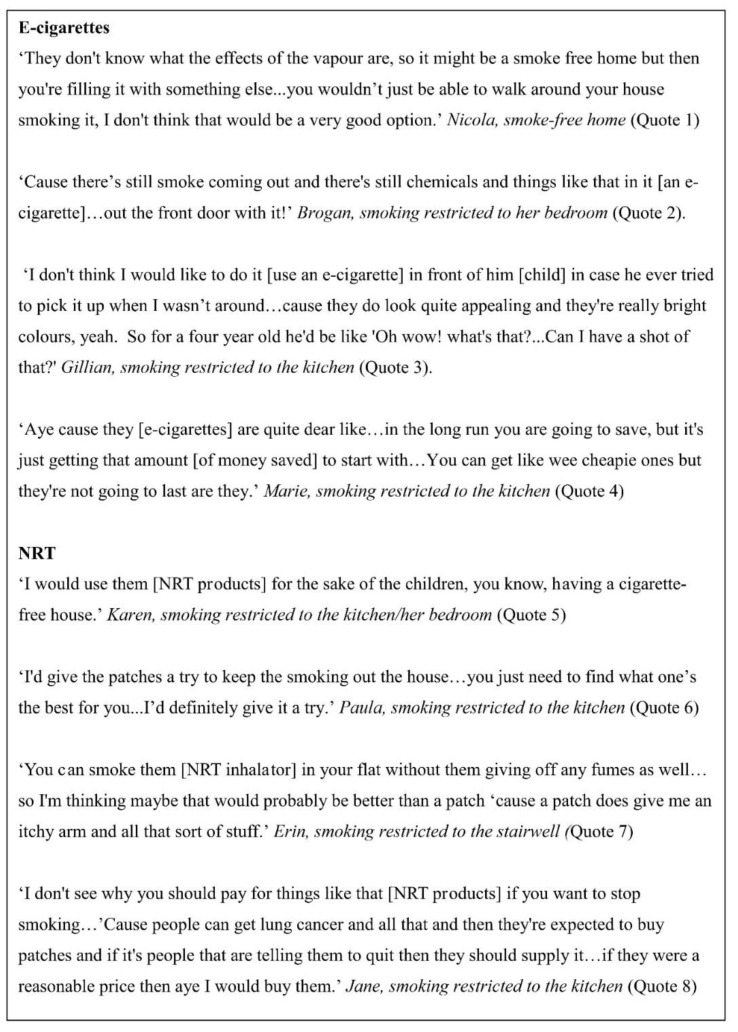
Parents’ views on using e-cigarettes and nicotine replacement therapy (NRT) to create a smoke-free home.

**Figure 3 ijerph-17-07305-f003:**
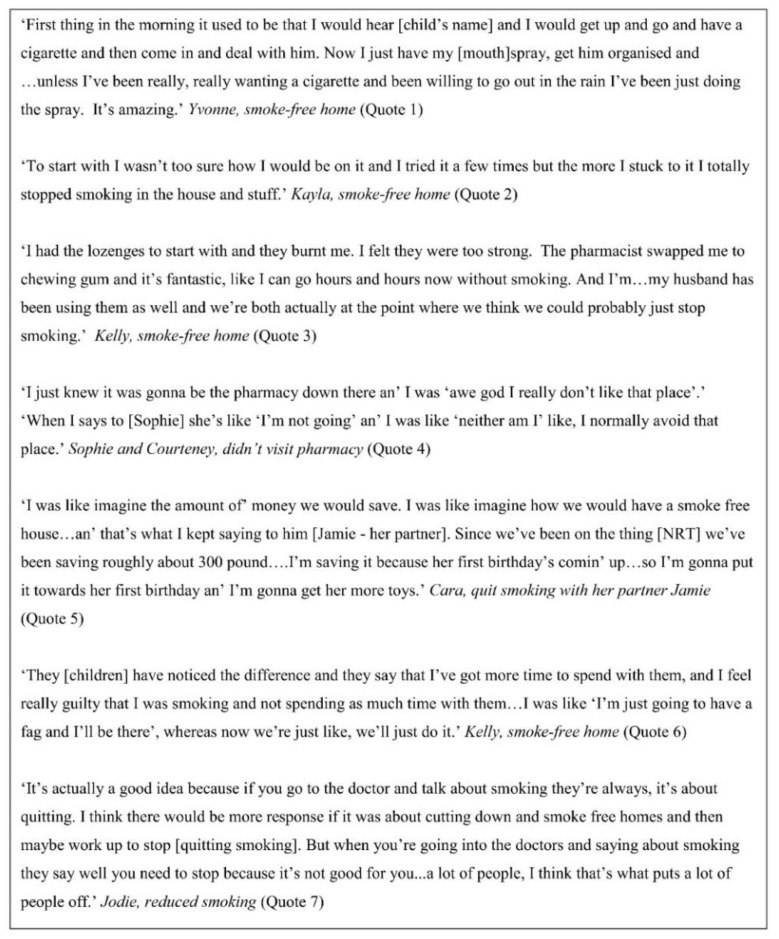
Parents’ experiences of using nicotine replacement therapy (NRT) to create a smoke-free home.

**Table 1 ijerph-17-07305-t001:** **Nicotine replacement therapy** (NRT) uptake and home-smoking rules pre and post 12 week NRT period.

Participant/Pseudonym	NRT Product Prescribed	Reported No. of Cigarettes Smoked Per day		Home-Smoking Rules	
		Pre-Intervention	Post-Intervention	Pre-Intervention	Post-Intervention
12—Lorraine	Gum	20+	5–10	Bathroom	Bathroom
15—Kayla	Gum	10	4–5	Hallway	Smoke-free home
16—Cara	Gum	2–3 (roll ups)	0	At the front door, door open	Quit smoking
2—Helen	Inhalator	8–9 (roll-ups)	8–9 (roll ups)	Kitchen/living room	Kitchen/living room
4—Ailsa	Inhalator	20	10	Kitchen	Smoke-free home
5—Cath	Inhalator	15–20	10–15	Kitchen	Kitchen
7—Karen	Inhalator	20	2	Unrestricted smoking in the home when no children present. Bedroom and kitchen when children present	Smoke-free home
10—Michelle	Inhalator	10	5–6	Balcony, lounge occasionally	Balcony, lounge occasionally
3—Kelly	Lozenges	20 (roll ups)	6 (roll ups)	Bathroom, laundry room	Smoke-free home
17—Jamie	Lozenges	8–9 (roll ups)	0	At the front door, door open	Quit smoking
9—Aileen	Inhalator, then switched to lozenges	40	20	Lounge (when children in bed) or kitchen	Kitchen
1—Jodie	Mouth spray	30–40	10–15	At the front door, door open	At the front door, door open
11—Marie	Mouth spray	20+	10–15	Bedroom, or in lounge when kids asleep	Smoke-free home
6—Terri	Gum, then mouth spray	15	10	Bedroom and Kitchen	Kitchen and outdoors
8—Yvonne	Lozenges, then switched to mouth spray	15–20	10–15	In the kitchen or at the front door, door open	Smoke-free home
13—Sophie	Did not visit pharmacy	15	15	At the back door or in the kitchen if raining	
14—Courteney	Did not visit pharmacy	15	15	At the back door, with the door open sometimes	

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
