# Peer review of "Supporting Parents Living in Disadvantaged Areas of Edinburgh to Create a Smoke-Free Home Using Nicotine Replacement Therapy (NRT): A Two-Phase Qualitative Study"

_ijerph, 2020, doi:10.3390/ijerph17197305_

Round 1
Reviewer 1 Report
Dear authors
The research design is well constructed and organized.
Would recommend that the keywords are MeSH terms, ate least three selected keywords are not Mesh terms.
Regarding the aims, as a rule, scientific articles are not written in this way, but in plain text.
Concerning the content and conclusions are very good.
Author Response
We are extremely grateful for your time and effort in reviewing our manuscript. Our responses are outlined below.
Point 1: The research design is well constructed and organized.
Point 4: Concerning the content and conclusions are very good.
Response 1: Thank you for your positive comments regarding our research design, content and conclusions are well constructed and organised.
Point 2: Would recommend that the keywords are MeSH terms, ate least three selected keywords are not Mesh terms.
Response 2: Thanks for your suggestion regarding MeSH terms. We used the ‘MeSH on Demand’ online function to identify alternative keywords we could include and have updated our original keywords on this basis. , with the exception of ‘smoke-free homes’ and ‘nicotine replacement therapy’. We have retained ‘smoke-free homes’ and ‘nicotine replacement therapy’ as whilst they are not listed as MeSH terms, they are specific to the article, and common terms within this field of research, and their inclusion is therefore in line with the IJERPH manuscript submission guidance.
Point 3: Regarding the aims, as a rule, scientific articles are not written in this way, but in plain text.
Response 3: We have edited the wording of Aim 2 (lines 113-116) to ensure it is easier to read. The aims now read as follows:
- Explore disadvantaged parents’ views and preferences regarding harm reduction approaches to creating a SFH (Phase 1)
- Pilot test a SFHs intervention based on parents’ preferences to use NRT to create a smoke-free home (Phase 2).
All acronyms used have been previously defined in text within the introduction.
Reviewer 2 Report
This study examines ways to help disadvanteged smokers with small children reduce consumption of quit altogether. It is a preliminary study with no control group and a small N. The merit in the study is the target population and an attempt to gather preference information for population members. It was interesting the e-Cigarettes were widely dismissed as an NRT alternative. Most reasons offered were not based on empirical evidence. Prior studies show that most long-term vapers will die of something else besides vaping-related causes (e.g., The Cochrane Collaboration 2016 meta-analysis headed by Hartman-Boyce.) I am aware of no definitive study showing vaping second-hand e-Cigarette vapor is harmful even though cheaply-made devices can leach toxins. Regardless, NRT seemed to work well and be well-accepted. Table 1 should be sorted by NRT method to help readers view outcomes comparatively by ingestion method.
Author Response
We are extremely grateful for your time and effort in reviewing our manuscript. Our responses are outlined below.
Point 1: This study examines ways to help disadvantaged smokers with small children reduce consumption of quit altogether. It is a preliminary study with no control group and a small N.
Response 1: We would like to clarify that this exploratory, qualitative study examines ways to help disadvantaged parents to create a smoke-free home. We have made this clearer in our study aims to avoid any potential confusion.
Point 2: The merit in the study is the target population and an attempt to gather preference information for population members. It was interesting the e-cigarettes were widely dismissed as an NRT alternative. Most reasons offered were not based on empirical evidence. Prior studies show that most long-term vapers will die of something else besides vaping-related causes (e.g., The Cochrane Collaboration 2016 meta-analysis headed by Hartman-Boyce.) I am aware of no definitive study showing vaping second-hand e-cigarette vapor is harmful even though cheaply-made devices can leach toxins. Regardless, NRT seemed to work well and be well-accepted.
Response 2: Thank you for your positive comments. We agree that parents’ views on the use of e-cigarettes to create a smoke-free home were interesting. Whilst you note that the reasons offered were not based on empirical evidence, the concerns that parents raised are similar to those identified in a previous research study exploring the notional use of e-cigarettes to create a SFH in a disadvantaged population of parents (see lines 240-247), suggesting these concerns may be shared by other parents in similar circumstances. This is an area that we feel warrants further research on this basis.
Point 3: Table 1 should be sorted by NRT method to help readers view outcomes comparatively by ingestion method.
Response 3: Thank you for your suggestion, which we have incorporated in our revisions (see line 272)
Reviewer 3 Report
Overall
This is an important topic given that mostly the smoke-free public policies do not cover homes as they are considered private domains. However, efforts to encourage adoption of smoke-free rules in homes are critical to fill that gap. One thing that seem not to come out clearly in this paper is whether this intervention was presented as an approach to encourage adoption of smoke-free rules in homes or was mainly presented with a focus on cessation. It might be useful to provide more information on what was presented to the participants. Was it targeted at smokers to quit, which, essentially would reduce SHS exposure at home. This may reflect the presentation of the intervention to allow for acceptability.
For example, in the methods section, the paper indicates that “The study’s National Health Service (NHS) cessation adviser (LS) then contacted each individual to arrange a suitable time to visit them either at home or at their EYC, to discuss study participation, obtain informed consent, and discuss NRT product choice.”
Cessation intervention provide an opportunity for smokers to quit. However, the research could have considered an approach to encouraging smoking bans in homes which leads to reduced SHS exposure, less smoking, and increasing attempts to quit – thus a combination of cessation support and home smoke bans. For example, mailing and coaching call approach used in a randomized trial in the USA (Kegler MC, Bundy L, Haardörfer R, et al. A minimal intervention to promote smoke-free homes among 2-1-1 callers: a randomized controlled trial. Am J Public Health. 2015 Mar;105(3):530-7. doi: 10.2105/AJPH.2014.302260. Epub 2015 Jan 20. PMID: 25602863; PMCID: PMC4330868).
It is not clear on the number of people who lived in a home that smoked. Did the intervention target all smokers in a home or only the person in the project? For example, in the methods section, the paper says “Individuals were eligible to take part if they (1) smoked in the home”. Thus, if they smoke but not in the home, they were excluded or if they lived with someone who smokes in the home, they were excluded.
One major challenge/drawback is the limited sample. It is a bit challenging to draw any significant generalization to a larger population.
Pre and post – the project might have considered interviewing the participants before the start of the program (baseline) and after completion. For example, in the methods section it is indicated that “Qualitative interviews were conducted with parents to capture their perspectives and 184 experiences of taking part in the pilot intervention. Interviews with parents took place once they had stopped using NRT for temporary abstinence, up to 12 weeks after their initial pharmacy prescription. Parents were also invited to take part in an interview if they had a) decided against 187 trying NRT after it had been prescribed; b) stopped using the NRT product before the prescription 188 period ended; or c) quit smoking before the end of the prescription period.”
Definitions
- May consider defining temporary abstinence.
- Quit
Page 7 lines 237 -238
“… parents living in disadvantage, …” is the sentence missing a word
Page 7 lines 243-245
Health-related concerns based on emerging evidence have recently been published, suggesting a number of ways in which, with respect to early-life exposures and health, e-cigarettes could present a cause for concern [28].
The sentence needs more clarification.
On this basis, in Phase 2, our intervention explored whether NRT 251 provision for use in the home supported parents to better protect children from SHS in 252 socioeconomically disadvantaged homes.
Page 5 lines 187 -
Parents were also invited to take part in an interview if they had a) decided against 187 trying NRT after it had been prescribed; b) stopped using the NRT product before the prescription period ended; or c) quit smoking before the end of the prescription period.
What topics were covered in the interviews/qualitative interviews
Individual qualitative interviews (N=7) were also conducted with FNP and pharmacy staff, and with the NHS adviser to ascertain their views on the feasibility and acceptability of this approach
Author Response
We are extremely grateful for your time and effort in reviewing our manuscript. Our responses are outlined below.
Point 1: This is an important topic given that mostly the smoke-free public policies do not cover homes as they are considered private domains. However, efforts to encourage adoption of smoke-free rules in homes are critical to fill that gap. One thing that seem not to come out clearly in this paper is whether this intervention was presented as an approach to encourage adoption of smoke-free rules in homes or was mainly presented with a focus on cessation. It might be useful to provide more information on what was presented to the participants. Was it targeted at smokers to quit, which, essentially would reduce SHS exposure at home. This may reflect the presentation of the intervention to allow for acceptability.
For example, in the methods section, the paper indicates that “The study’s National Health Service (NHS) cessation adviser (LS) then contacted each individual to arrange a suitable time to visit them either at home or at their EYC, to discuss study participation, obtain informed consent, and discuss NRT product choice.”
Cessation intervention provide an opportunity for smokers to quit. However, the research could have considered an approach to encouraging smoking bans in homes which leads to reduced SHS exposure, less smoking, and increasing attempts to quit – thus a combination of cessation support and home smoke bans. For example, mailing and coaching call approach used in a randomized trial in the USA (Kegler MC, Bundy L, Haardörfer R, et al. A minimal intervention to promote smoke-free homes among 2-1-1 callers: a randomized controlled trial. Am J Public Health. 2015 Mar;105(3):530-7. doi: 10.2105/AJPH.2014.302260. Epub 2015 Jan 20. PMID: 25602863; PMCID: PMC4330868).
Response 1: Many thanks for your comments. We have revised the wording of our second research aim (lines 113-116) which now more clearly states that this two phase study is focussed on encouraging parents to create a smoke-free home. We have also added text to clarify that when participants were recruited for Phase 2 of the study, the focus of the study on creating a smoke-free home was clarified (lines 170-172). This was important given that NRT is more usually associated with quit attempts. We appreciate that the involvement of an National Health Service (NHS) cessation adviser (LS), who visited each participant in their home to discuss NRT product options with them (Phase 2) may lead to some confusion about whether the study is focused on smoke-free homes, quitting smoking, or both of these outcomes. On this basis, we have removed the term ‘cessation’ from line 173, and refer to our NHS collaborator as an NHS adviser throughout the paper. We hope that collectively, these revisions help to ensure there is no confusion regarding the focus of the study on supporting parents to create a smoke-free home. We agree that it would be interesting to conduct research in future which links cessation and smoke-free homes, especially given recent research (lines 46-47) which has shown that smokers who introduce smoke-free home rules are more likely to quit in the following six months.
Point 2: It is not clear on the number of people who lived in a home that smoked. Did the intervention target all smokers in a home or only the person in the project? For example, in the methods section, the paper says “Individuals were eligible to take part if they (1) smoked in the home”. Thus, if they smoke but not in the home, they were excluded or if they lived with someone who smokes in the home, they were excluded.
Response 2: We collected data at the start of each interview on the number of people living in each home, and whether other adult household members smoked – this is not presented in the article as our intervention was conducted with individuals, and not at household level. We did find, as outlined in lines 350-352, a few participants informally shared their NRT with their partner at home, during the course of the study. Whilst this was not formally encouraged, this approach fits with recent calls for future SFH interventions to be delivered at the household level, rather than being largely mother-led. In terms of our inclusion criteria, as stated in line 160, individuals were eligible to take part if they smoked in the home. On that basis, as you say, if individuals smoked only outside of the home, or did not smoke themselves, they were not eligible to take part.
Point 3: One major challenge/drawback is the limited sample. It is a bit challenging to draw any significant generalization to a larger population.
Response 3: The goal of most qualitative studies is not to generalise to a larger population, but rather to provide a rich, contextualized and in-depth understanding of the topic being studied. This exploratory qualitative work provides a detailed initial understanding of the ways in which NRT could be used to assist parents to create a smoke-free home, the inference being that this approach could be used effectively in other populations and settings. Our findings will inform future research to assess the feasibility and cost effectiveness of this approach, at which point a larger sample size will be totally appropriate. However, for this qualitative study, our sample size was agreed in order to include a range of personal contexts, whilst enabling data collection and analysis in the time available. The experience of most qualitative researchers conducting interviews is that, little new information comes out of transcripts once you have analysed 15 or so (Green & Thorogood, 2018), if you have a relatively homogenous group of participants, as ours was. On this basis, we are content that our sample size was adequate for the purpose of our study.
Point 4: Pre and post – the project might have considered interviewing the participants before the start of the program (baseline) and after completion. For example, in the methods section it is indicated that “Qualitative interviews were conducted with parents to capture their perspectives and experiences of taking part in the pilot intervention. Interviews with parents took place once they had stopped using NRT for temporary abstinence, up to 12 weeks after their initial pharmacy prescription. Parents were also invited to take part in an interview if they had a) decided against trying NRT after it had been prescribed; b) stopped using the NRT product before the prescription period ended; or c) quit smoking before the end of the prescription period.”
Response 4: Thank you for your suggestion. A pre and post design would be preferable for a quantitative study to demonstrate intervention effects, but we feel this approach is less well suited to our exploratory qualitative study. Our main focus in the interviews was learning about participant experiences of using NRT in this way, their experience of engaging with our NHS adviser and pharmacy staff. None of this information could have been gained from interviews conducted prior to NRT use. On this basis, we feel that one interview at the end of NRT use was sufficient in this study, and this also ensured that participants were not over-burdened, given qualitative interviews can be quite lengthy.
Point 5: Definitions - May consider defining temporary abstinence.
Response 5: We have now defined temporary abstinence in lines 74-76.
Point 6: Page 7 lines 237 -238 “… parents living in disadvantage, …” is the sentence missing a word
Response 6: We have amended line 246, and lines 35-36, which both now read ‘parents living in disadvantaged circumstances’.
Point 7: Page 7 lines 243-245
Health-related concerns based on emerging evidence have recently been published, suggesting a number of ways in which, with respect to early-life exposures and health, e-cigarettes could present a cause for concern [28].
The sentence needs more clarification.
Response 7: We have added additional detail to clarify this point in lines 252-257.
Point 8: Page 5 lines 187 -
Parents were also invited to take part in an interview if they had a) decided against trying NRT after it had been prescribed; b) stopped using the NRT product before the prescription period ended; or c) quit smoking before the end of the prescription period.
What topics were covered in the interviews/qualitative interviews?
Response 8: We have added a sentence (lines 191-194) to clarify the key topics covered in the Phase 2 topic guide. Thank you for highlighting this.